# SUMO-mediated regulation of NLRP3 modulates inflammasome activity

Rachael Barry [1,2], Sidonie Wicky John[1], Gianmaria Liccardi[1], Tencho Tenev[1], Isabel Jaco[1], Chih-Hong Chen[3], Justin Choi[3], Paulina Kasperkiewicz[4], Teresa Fernandes-Alnemri[5], Emad Alnemri[5], Marcin Drag[4], Yuan Chen[3] & Pascal Meier[1]

The NLRP3 inflammasome responds to infection and tissue damage, and rapidly escalates the intensity of inflammation by activating interleukin (IL)-1β, IL-18 and cell death by pyroptosis. How the NLRP3 inflammasome is negatively regulated is poorly understood. Here we show that NLRP3 inflammasome activation is suppressed by sumoylation. NLRP3 is sumoylated by the SUMO E3-ligase MAPL, and stimulation-dependent NLRP3 desumoylation by the SUMO-specific proteases SENP6 and SENP7 promotes NLRP3 activation. Defective NLRP3 sumoylation, either by NLRP3 mutation of SUMO acceptor lysines or depletion of MAPL, results in enhanced caspase-1 activation and IL-1β release. Conversely, depletion of SENP7 suppresses NLRP3-dependent ASC oligomerisation, caspase-1 activation and IL-1β release. These data indicate that sumoylation of NLRP3 restrains inflammasome activation, and identify SUMO proteases as potential drug targets for the treatment of inflammatory diseases.

[1] The Breast Cancer Now Toby Robins Research Centre, Institute of Cancer Research, Mary-Jean Mitchell Green Building, Chester Beatty Laboratories, 237 Fulham Road, London SW3 6JB, UK. [2] MRC Centre for Molecular Bacteriology and Infection, Department of Life Sciences, Imperial College London, London SW7 2AZ, UK. [3] Department of Molecular Medicine, Beckman Research Institute of City of Hope, Duarte, California 91010, USA. [4] Division of Bioorganic Chemistry, Department of Chemistry, Wroclaw University of Technology, Wyb. Wyspianskiego 27, 50-370 Wroclaw, Poland. [5] Department of Biochemistry and Molecular Biology, Kimmel Cancer Center, Thomas Jefferson University, Philadelphia 19107 PA, USA. Correspondence and requests for materials should be addressed to P.M. (email: pmeier@icr.ac.uk)

nflammasomes are key signalling complexes of the innate immune system that drive activation of caspase-1 in response to microbial and non-microbial danger signals[1], including pathogen-derived proteins, lipids, nucleic acids, polysaccharides, crystalline materials, extracellular ATP, specific host proteins, and alterations in ion levels and osmolarity. Activation of caspase-1 results in the cleavage and maturation of the highly inflammatory cytokines interleukin (IL)-1β and IL-18[2]. In addition, caspase-1 also cleaves gasdermin-D, which triggers programmed inflammatory cell death (pyroptosis) that enables the release of IL-1β, IL-18 and other DAMPs[3,4]. Tight regulation of inflammasome activation is crucial as aberrant or excessive activation of caspase-1 is associated with various diseases including gout disease, type-II diabetes, Alzheimer's disease and atherosclerosis[5–10]. Accordingly, genetic mutations that result in uncontrolled activation of the NLRP3 inflammasome are linked to cryopyrinopathies or cryopyrin-associated periodic fever syndromes (CAPS)[5].

Upon exposure to exogenous or endogenous stimuli, NLRP3 assembles a canonical multimeric inflammasome complex comprising the adaptor Apoptosis-associated speck-like protein containing a CARD (ASC) and the effector pro-caspase-1 to mediate the activation of caspase-1[11]. At present, there is no evidence of direct ligand binding by NLRP3, which led to the hypothesis that NLRP3 senses changes in the cellular milieu. Numerous activation models for the NLRP3 inflammasome have been proposed including lysosomal rupture, mitochondrial damage, ROS production, potassium efflux and plasma membrane rupture (reviewed in[12,13]). However, despite considerable efforts, the precise mechanism by which NLRP3 senses these cellular changes remains unclear.

NLRP3 is expressed by myeloid cells and is up-regulated in response to the stimulation of macrophages with pathogen-associated molecule patterns (PAMPs)[14]. A two-signal model has emerged for NLRP3/ASC/caspase-1-mediated IL-1β maturation, whereby PAMPs, such as LPS, provide 'signal I' and diverse agents (such as ATP and nigericin) act as 'signal II'. Generally, signal I induces the transcriptional expression of NLRP3 and IL-1β. Additionally, signal I can non-transcriptionally prime NLRP3 by stimulating its deubiquitylation[15]. This process is dependent on mitochondrial ROS and can be inhibited by antioxidants. Signal II induces processing of pro-IL-1β to the p17 active form, which is then released into the extracellular space.

In contrast to other inflammasomes, the NLRP3 inflammasome is activated in response to a wide variety of stimuli including pore-forming toxins, nigericin, maitotoxin, and valinomycin, as well as the P2X7 channel activator ATP, uric acid, silica crystals, and the widely used adjuvant aluminium hydroxide[1]. A mechanism for NLRP3 inflammasome activation that unites all signal II stimuli has yet to emerge. One intriguing possibility is that the cellular changes sensed by NLRP3 may converge on a common regulatory mechanism, for example modulation of post-translational modifications (PTMs), such as phosphorylation, ubiquitylation or sumoylation. Protein modification by small ubiquitin-like modifier (SUMO) allows the dynamic regulation of proteins as most substrates undergo a constant turnover of SUMO conjugation and de-conjugation[16]. Similar to ubiquitylation, sumoylation is regulated by a specialised set of activating (E1), conjugating (E2) and ligating (E3) enzymes, and is reversed by specific isopeptidases referred to as sentrin/SUMO-specific proteases (SENPs)[16,17]. The three SUMO proteins (SUMO-1, SUMO-2 and SUMO-3) can be covalently conjugated to proteins as a single moiety (SUMO-1) or as polymeric SUMO chains (SUMO-2 and SUMO-3). Sumoylation is essential for maintaining cell homeostasis, and as such is implicated in many cellular processes including cellular stress response, DNA replication and repair, apoptosis and inflammation[18].

Although the majority of sumoylated proteins are localised to the nucleus, a number of cytoplasmic, mitochondrial and membrane-associated targets have recently been identified[19–21]. Sumoylation of substrates preferentially occurs on a lysine residue in the canonical SUMO consensus motif ψKx(D/E), in which ψ is a large hydrophobic residue and x is any amino acid followed by an acidic residue[16,22]. The hydrophobic and acidic residues stabilise the interaction between the substrate and the E2 enzyme, UBC9[23].

Here we demonstrate that conjugation of SUMO negatively regulates the NLRP3 inflammasome, implicating sumoylation as a fundamental post-translational mediator of innate immune signalling. We show that NLRP3 is sumoylated at multiple sites, and identify MAPL as a SUMO E3 ligase responsible for NLRP3 sumoylation. Accordingly, depletion of MAPL in mouse bone marrow-derived macrophages (BMDMs) significantly enhances NLRP3 inflammasome formation and activity. In addition, using nuclear magnetic resonance (NMR) and in vitro sumoylation assays we identify lysine (K) 689 as a sumoylation site of NLRP3. Mutation of this K689 residue results in hyper-activation of NLRP3. Furthermore, we show that desumoylation of NLRP3 by the desumoylase enzymes SENP6 and SENP7 controls NLRP3 inflammasome activation in BMDMs. Consistent with the notion that SENP-mediated desumoylation contributes to full activation of the NLRP3 inflammasome, we find that depletion of these SENPs attenuates inflammasome formation and downstream signalling. Together our data demonstrate that NLRP3 is regulated in a SUMO-dependent manner, and that desumoylation of NLRP3 by SENP6 and SENP7 contributes to full inflammasome activation.

## Results

**NLRP3 is sumoylated in vivo.** Deregulated activation of NLRP3 is associated with auto-inflammatory disorders characterised with excessive production of IL-1β[24]. It is presently unclear how the NLRP3 inflammasome is kept silent under base-line conditions. Bioinformatics analysis of NLRP3 using four independent computational programmes to detect sumoylation sites (SUMO-plot™ Analysis Programme (Abgent), JASSA[25], GPS-SUMO[26], and Ron Hay's SUMO motif search tool) identified six potential SUMO-conjugation consensus motifs, five of which are evolutionarily conserved (red boxes) from mouse to man (Fig. 1a, b). In contrast to NLRP3, no such evolutionarily conserved SUMO consensus motifs were identified in ASC and pro-caspase-1. In addition to SUMO consensus motifs, we also identified a putative SUMO-interaction motif (SIM) within the LRR of mouse (Q8R4B8, SIM: amino acids 797–800 (LVEL) and human NLRP3 (Q96P20, SIM: amino acids 800–803 (LVEL)). This was the only putative SIM that was predicted by both JASSA and GPS algorithms. Although this putative SIM is evolutionarily conserved it has a relatively low probability score.

To test whether NLRP3 is regulated by sumoylation in vivo, we immunoprecipitated NLRP3 from cellular extracts following denaturing conditions, avoiding the isolation of protein complexes. We used the previously described Nlrp3$^{-/-}$ BMDMs that re-express NLRP3 from a transgene (Nlrp3$^{-/-.reconNLRP3}$, subsequently referred to as N1-8)[15]. The N1-8 macrophages express FLAG-tagged NLRP3 at a level comparable to that present in un-induced wild-type (WT) macrophages (basal level)[15]. Immunoblotting of the NLRP3 immunoprecipitates using SUMO-2/-3 specific antibodies identified high apparent molecular weight NLRP3 species corresponding to sumoylated NLRP3 (Fig. 1c). No such signals were detected in immunoprecipitates of Nlrp3$^{-/-}$ BMDMs, indicating that the detected bands were specific for sumoylated NLRP3. To test whether NLRP3 is also sumoylated

in human cells, we transfected human NLRP3 into HEK293T cells, and conjugation of endogenous SUMO-2/-3 to NLRP3 was confirmed by immunoprecipitation of NLRP3 (Fig. 1d). Treatment with the non-selective isopeptidase inhibitor PR-619, which also blocks SUMO deconjugating enzymes, significantly enhanced the appearance of sumoylated NLPR3 (Fig. 1d, compare lanes 3 with 4). Conjugation of SUMO-2/-3 to human NLRP3 was also confirmed by reciprocal immunoprecipitation of HA-tagged WT SUMO-3 or a mutant form of SUMO-3 (SUMO-3$^{\Delta GG}$) that cannot be conjugated to substrates due to deletion of the critical C-terminal di-Gly motif of SUMO-3 (Fig. 1e). Under these conditions, NLRP3

was readily immunoprecipitated with WT SUMO-3, but not with SUMO-3$^{\Delta GG}$ (Fig. 1e), demonstrating the selectivity of the purification and SUMO conjugation. Consistent with a mono-sumoylation event, we observed NLRP3 species that migrated approximately 10 kDa above unmodified NLRP3. In addition to mono-sumoylated NLRP3, higher molecular weight smears of NLRP3 were also observed, indicating the presence of polymeric SUMO-3 chains or co-modification with SUMO and ubiquitin (Ub) chains (Fig. 1e, lane 3). Taken together, these results indicate that NLRP3 is sumoylated in murine macrophages and human cells.

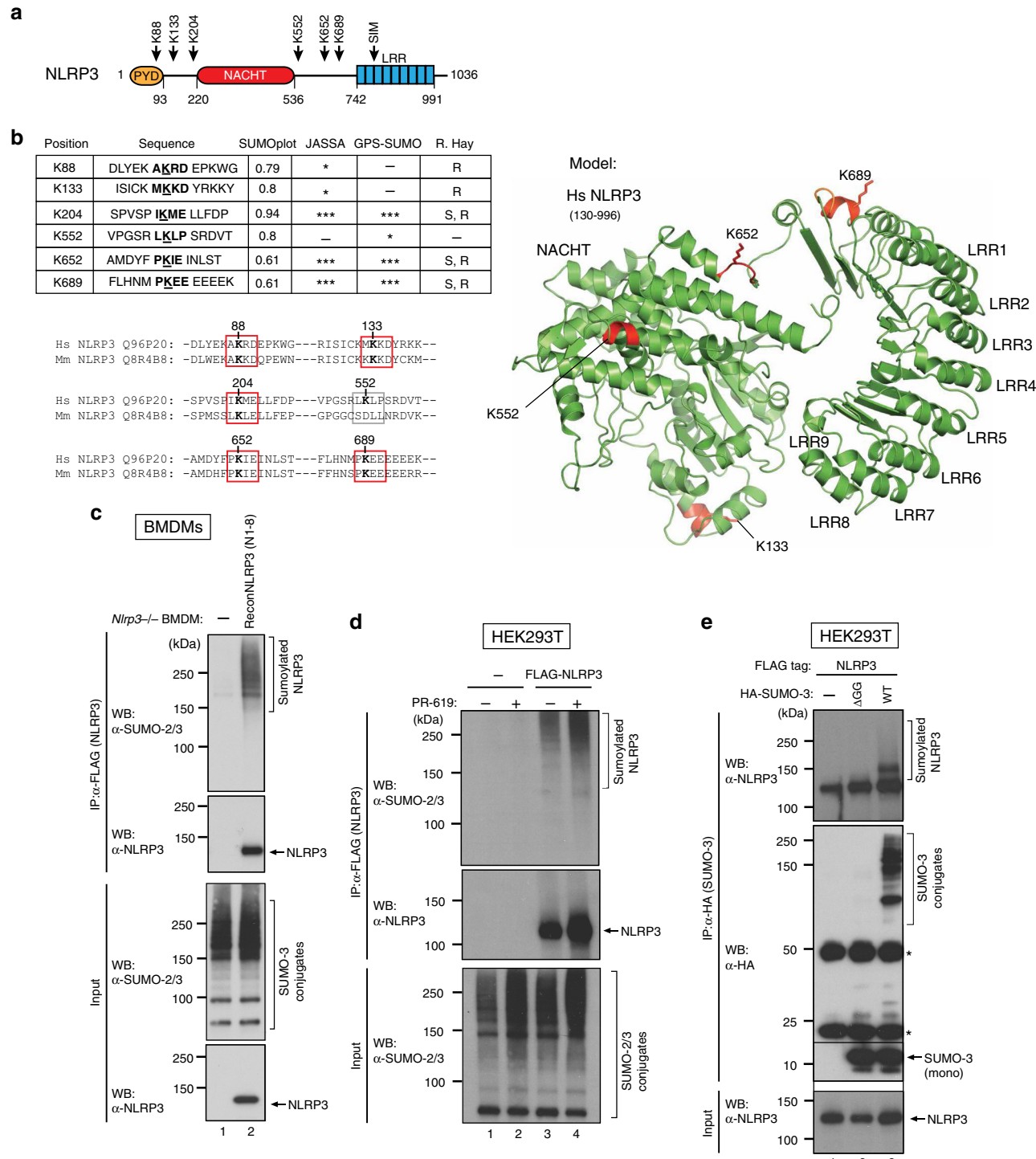

**Signal II triggers loss of sumoylation of NLRP3.** Since NLRP3 is modified by SUMO under unstimulated conditions we investigated whether NLRP3-activating stimuli (signal II) alters the sumoylation state of NLRP3. Accordingly, we found that treatment with the bacterial toxin nigericin induced notable reduction of sumoylated NLRP3 (Fig. 2a). Under the same conditions, NLRP3 was also less ubiquitylated, as previously reported[15], suggesting that the activation signal triggers reduction in both ubiquitylated and sumoylated NLRP3. Of note, nigericin-mediated reduction in sumoylation of NLRP3 was selective as other sumoylated proteins, such as RanGAP1, remained sumoylated under the experimental conditions (Fig. 2a). Moreover, there was no apparent change in the overall smearing pattern of the total SUMO-2/-3 proteome, ruling out the possibility that nigericin stimulates a reduction in sumoylation non-selectively.

We next determined the sumoylation status of endogenous NLRP3 in WT BMDMs. To visualise the interaction between endogenous NLRP3 and SUMO-2/-3 in intact WT BMDMs, we applied in situ proximity ligation assay (PLA). In addition to detecting protein-protein interactions[27] and inflammasome activation[28–30], this method has been described for the study of PTMs, specifically sumoylation[31]. Using primary antibodies against NLRP3 and SUMO-2/-3, which in turn were recognised by oligonucleotide-coupled secondary antibodies, we obtained discrete proximity labelling of NLRP3 and SUMO-2/-3 under unstimulated conditions (Fig. 2b, upper panel). Likewise, distinct NLRP3-SUMO-2/-3 PLA signals were detected upon LPS priming, which were comparable to the ones of unstimulated WT BMDMs (Fig. 2b). Of note, priming with LPS did not influence NLRP3 sumoylation (Supplementary Fig. 1a). Importantly, no such PLA signals were observed in *Nlrp3*[−/−] control BMDMs, demonstrating the suitability of the PLA to detect NLRP3-SUMO-2-/3 interactions. Interestingly, treatment with nigericin and ATP significantly reduced proximity labelling of NLRP3 and SUMO-2/-3, which is entirely consistent with our biochemical analysis (Fig. 2a). While proximity labelling of NLRP3 and SUMO-2/-3 was markedly reduced, general expression of NLRP3 and SUMO-2/-3 was unaffected by the addition of LPS and nigericin, as determined by immunofluorescence staining of the individual proteins (Supplementary Fig. 1b). As expected, PLA with antibodies against NLRP3 and the inflammasome adapter component ASC revealed that inflammasome complexes were absent under base-line conditions, and specifically formed upon treatment with LPS and nigericin or ATP (Fig. 2b and Supplementary Fig. 1c). Together, our data indicate that inflammasome formation occurs in concordance with loss of sumoylation of NLRP3, supporting the working hypothesis that sumoylation impedes NLRP3 complex formation.

**MAPL suppresses inflammasome activation.** To study SUMO-mediated regulation of NLRP3, we next identified a SUMO E3 ligase that modifies NLRP3. While the SUMO E2 enzyme can directly interact with a SUMO consensus motif present in substrates, this interaction is insufficient for an efficient SUMO transfer in vivo and needs to be stabilised either by additional E2 interactions or by E3 ligases[16]. Particularly, E3 ligases interact with the substrate and the charged E2 enzyme to catalyse the discharge of the thioester-bound SUMO from the E2 enzyme to the substrate. Although the majority of SUMO E3 ligases identified are nuclear localised, a mitochondria-localised SUMO E3 ligase has recently been identified, mitochondrial-anchored protein ligase (MAPL/MUL1). MAPL is known to play a role in the control of mitochondria morphology and localisation[32,33]. In addition, MAPL is involved in the modulation of innate immune defence against viruses by sumoylating and inhibiting DDX58[34,35]. Given that NLRP3 can also localise to mitochondria[36–38], we tested the ability of MAPL to bind to and sumoylate NLRP3. Reciprocal immunoprecipitation assays revealed that NLRP3 interacted with MAPL (Fig. 3a, b). Consistent with the notion that MAPL can function as the SUMO-E3 ligase for NLRP3, we found that WT MAPL sumoylated NLRP3 (Fig. 3c). In contrast, a SUMO-E3 deficient MAPL mutant (MAPL[C399A]), carrying a point mutation in the zinc coordinating cysteine of the RING domain, failed to sumoylate NLRP3. These results indicate that MAPL sumoylates NLRP3 in a RING-finger dependent manner.

To test whether MAPL interacts with NLRP3 at endogenous levels, we applied PLA in WT BMDMs using primary antibodies against MAPL and NLRP3. In unstimulated as well as in LPS-treated WT BMDMs, we obtained discrete proximity labelling of MAPL and NLRP3 (Figs. 3d and 4a), corroborating the notion that the two proteins are in close proximity. Importantly, no such signals were observed in *Nlrp3*[−/−] BMDMs or upon knockdown of *Mapl* (Figs. 3d and 4a). Excitingly, treatment with nigericin almost completely abolished MAPL/NLRP3 proximity signals (Figs. 3d and 4a). This may be due, at least in part, to depletion of MAPL because nigericin treatment caused a decrease in MAPL staining, as visualised by conventional immunofluorescence (Supplementary Fig. 2). This suggests that inflammasome-activating stimuli disrupt the interaction between MAPL and NLRP3, thus ceasing MAPL-mediated sumoylation of NLRP3.

To test whether MAPL is indeed required for the association of endogenous SUMO and NLRP3, we depleted *Mapl* from WT BMDMs by RNAi. Although knockdown efficiency was approximately 50% (Supplementary Fig. 3a, b), a considerable reduction in MAPL/NLRP3 proximity signals was observed, which served as an additional control (Fig. 4a, top panel). RNAi-mediated depletion of *Mapl* dramatically reduced the association between NLRP3/SUMO-2/-3, as seen by the reduced proximity signal in unstimulated and LPS-treated WT BMDMs (Fig. 4a, middle panel).

Given the reduced association between NLRP3 and SUMO-2/-3 following *Mapl* depletion, we next tested whether depletion of *Mapl* enhanced inflammasome formation using PLA[28–30].

---

**Fig. 1** NLRP3 is modified by SUMO-2/-3. **a** Schematic representation of the six predicted SUMO-conjugation motifs (ψKXE) and SUMO-interaction motif (SIM) in the NLRP3 protein. **b** Left-hand side, details of the SUMO-conjugation motifs and summary of prediction software scores. Analysis using R. Hay website provides motif type (S = strict and R = relaxed). Conservation of the predicted SUMO conjugation motifs between mouse and human NLRP3 are depicted underneath. Red boxes indicate evolutionary conserved sites, grey boxes indicate sites that are only present in humans. Right-hand side, structural prediction of NLRP3 depicting the identified SUMO motifs. The acceptor lysines shown in (a) are highlighted as red sticks. Note, K88 and K204 are in flexible loops and hence are not part of the model. The structural prediction of NLRP3 was created using PHYRE2. **c** Sumoylation of NLRP3 in murine *Nlrp3*[−/−] BMDMs and N1-8 BMDMs that were reconstituted with FLAG-tagged NLRP3 (N1-8). NLRP3 was immunoprecipitated and the presence of the indicated proteins was evaluated using the indicated antibodies. For this and all subsequent immunoblots, representative immunoblots are shown from at least three independent experiments. **d** Sumoylation of NLRP3 in HEK293T cells expressing FLAG-tagged human NLRP3 in the presence and absence of PR-619. NLRP3 was immunoprecipitated and the presence of the indicated proteins was evaluated using the indicated antibodies. **e** Sumoylation of NLRP3 in HEK293T cells transfected with FLAG-tagged human NLRP3 and HA-SUMO-3 or SUMO3[ΔGG] (non-conjugatable) control. Proteins covalently bound to HA-SUMO-3 were purified under denaturing conditions using HA-resin. The presence of the indicated proteins was evaluated by immunoblotting

Consistently, we found that depletion of *Mapl* indeed enhanced oligomerisation of ASC (Supplementary Fig. 3c) and the formation of inflammasome complexes (Fig. 4a, lower panel). Of note, PLA detects hundreds of co-localisation dots[28–30] instead of the characteristic ASC speck. Because PLA is a PCR-based method, it is probably more sensitive than conventional antibody-based confocal microscopy, and hence may also detect pre-speck complexes.

Enhanced inflammasome formation upon *Mapl* deletion implies that the NLRP3 inflammasome will be more active. Therefore, we next defined the contribution of MAPL in regulating inflammasome activity. To rule out any transcriptional involvement of sumoylation in NLRP3 activation, we used the N1-8 BMDMs in which the transcriptional induction of *Nlrp3* does not depend on priming[15]. To further eliminate any transcriptional involvement, we used a short 10 min incubation

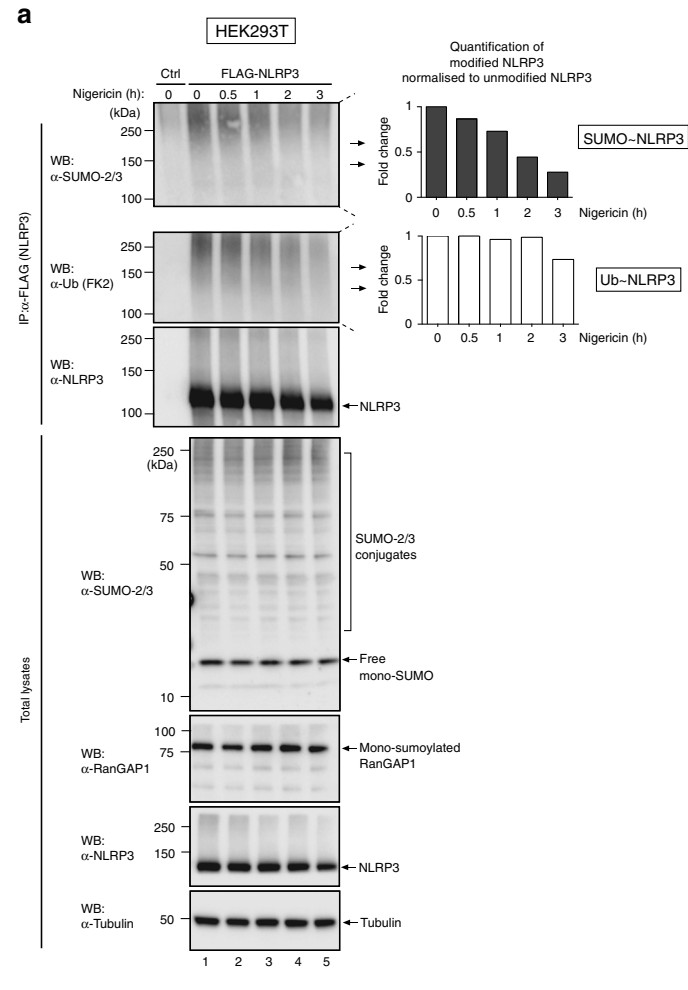

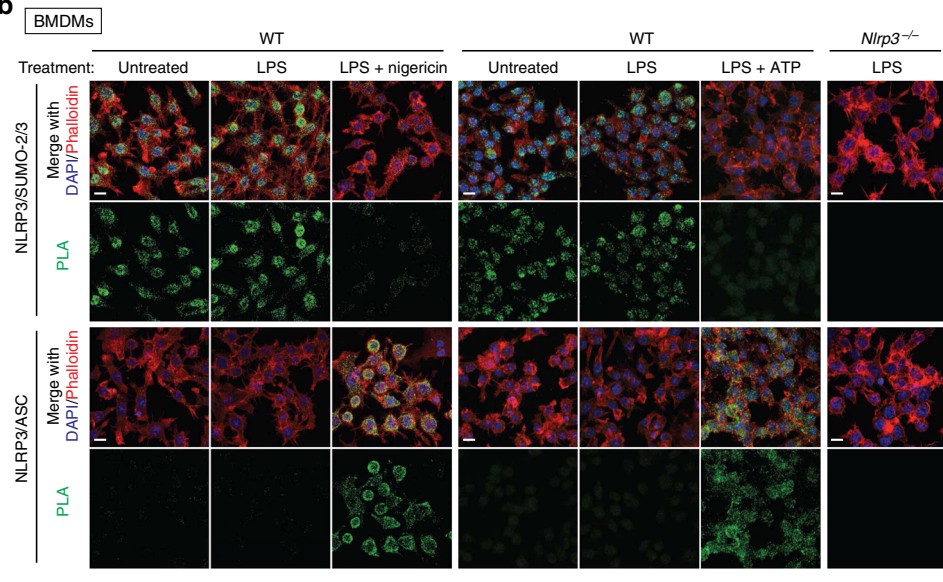

with LPS, which primes the NLRP3 inflammasome by a non-transcriptional mechanism[15] (Fig. 4b). In agreement with the notion that MAPL regulates NLRP3 under physiological conditions, we found that RNAi-mediated depletion of *Mapl* in N1-8 BMDMs significantly enhanced nigericin-induced activation of caspase-1 and production of IL-1β (Fig. 4c, d). In contrast, depletion of *Mapl* had no effect on AIM2-mediated activation of caspase-1 in response to cytoplasmic DNA (Fig. 4c, lanes 9–12). As knockdown of *Mapl* selectively affects the NLRP3 inflammasome, these results support the notion that NLRP3, but not AIM2 or common components such as ASC and caspase-1, is subject to SUMO-mediated regulation by MAPL.

**Mutation of sumoylated K residues causes hyperactive NLRP3.** To functionally test the role of the putative SUMO-conjugation consensus motifs of NLRP3, we mutated the surface exposed SUMO-acceptor lysines (K) of each motif and analysed the ability of these mutants to activate caspase-1. To this end, we made use of a previously established HEK293T cell line, stably expressing ASC and caspase-1 (293T-ASC-caspase-1 cells)[39]. As previously reported, expression of WT NLRP3 stimulated activation of caspase-1 in these cells (Fig. 5a–c). Intriguingly, we found that mutating individual SUMO conjugation consensus sites in isolation slightly enhanced the ability of NLRP3 to activate caspase-1, ranging from 1.3-fold to 1.7-fold when compared to WT NLRP3 (Fig. 5a). Mutating all six SUMO acceptor K of NLRP3 (NLRP3^6K>R) further enhanced activation of caspase-1 (Fig. 5b, c), while mutating the SIM (LVEL to AAEA) reproducibly lowered the ability of NLRP3 to activate caspase-1 (Fig. 5c). The potency of NLRP3^6K>R to activate caspase-1 was comparable to that of reported CAPS disease mutants, such as NLRP3^R260W, NLRP3^E690K and NLRP3^E692K (Fig. 5b, c and Supplementary Fig. 4a), which served as controls[40].

Residues of the SUMO consensus motifs interact directly with the SUMO E2 ligase, UBC9[23,41]. To test whether the predicted SUMO consensus motifs indeed interact with UBC9, we performed NMR with UBC9 and an NLRP3 peptide stretch containing the SUMO consensus motif surrounding K689. Increasing NMR chemical shift perturbation (CSP) was induced by increasing peptide:UBC9 molar ratios of the WT peptide (Fig. 5d). WT peptide induced significant CSP on residues L94, D127, A129 and Y134 (Fig. 5d). Substitution of the target K to R (K689R peptide) lead to weak CSP, indicating reduced binding to UBC9 (Fig. 5d). This result suggests that the binding mode is similar to that of other ψKx(D/E) motifs that we examined previously[42] (depicted model Supplementary Fig. 4b). We next performed an in vitro conjugation assay to determine if K689 can be modified by SUMO. As expected, the peptide corresponding to the WT NDSM was modified by SUMO-1 and SUMO-3, whereas the peptide containing the K > R substitution was not (Fig. 5e). Together, these data suggest that SUMO can be conjugated to K689 of NLRP3. Mutating K689 to R in full length NLRP3 (NLRP3^K689R), like mutating other SUMO conjugation consensus

sites in isolation, had no effect on the overall protein levels as well as sumoylation or ubiquitylation pattern of NLRP3 in vivo, likely due to additional modifications occurring at alternative sites (Supplementary Fig. 4c). This is in slight conflict to a previous in vitro assay, suggesting that a K689R NLRP3 mutant is less polyubiquitylated compared to WT NLRP3 (see Discussion)[43]. Of note, mutation of single residues reported in CAPS disease mutants (NLRP3^R260W, NLRP3^E690K and NLRP3^E692K) did not reduce the overall sumoylation smearing pattern on NLRP3 (Fig. 5f, and Supplementary Fig. 4d). However, consistent with the notion that NLRP3 is sumoylated at multiple sites, we found that substitution of all six K residues to R (NLRP3^6K>R) diminished NLRP3 sumoylation by 50% when compared to WT NLRP3 (Fig. 5f). Importantly, mutating these SUMO acceptor K residues did not affect overall ubiquitylation of NLRP3 (Fig. 5f and Supplementary Fig. 4c). Accordingly, WT NLRP3 and NLRP3^6K>R were ubiquitylated to similar extent. As sumoylation was not completely abrogated, our data suggest that alternative K residues can also serve as SUMO acceptors when the canonical sites are mutated. Conversely, mutation of the putative SIM of NLRP3 (NLRP3^SIM-mut) caused enhanced sumoylation of NLRP3 (Fig. 5f).

**SENP6 and 7 regulate NLRP3 activation and IL-1β secretion.** Next, we assessed the contribution of SUMO de-conjugating enzymes to the regulation of NLRP3. We found that RNAi-mediated knockdown of *Senp6* and *Senp7* reproducibly suppressed autocatalytic caspase-1 maturation and secretion in response to ATP (Fig. 6a). Under the same conditions, depletion of *Senp6* and *Senp7* had no effect on AIM2-mediated activation of caspase-1 in response to cytosolic DNA (Fig. 6a, lane 9–16). Partial depletion of *Senp6* and/or *Senp7* (~50% knockdown) also suppressed NLRP3-mediated activation and release of caspase-1 in response to nigericin in N1-8 as well as WT BMDMs (Fig. 6b, and Supplementary Fig. 5a–c). While partial RNAi-mediated depletion of *Senp6* and *Senp7* reduced ATP and nigericin-induced caspase-1 activation, knockdown of other members of the SENP protein family did not significantly modulate the ability of NLRP3 to activate caspase-1 (Fig. 6b, and Supplementary Fig. 5a). Consistent with the notion that SENP7 contributes to activation of the NLRP3-inflammasome, we found that depletion of *Senp7* reduced ASC oligomerisation, caspase-1 activation/release, as well as IL-1β processing/release (Fig. 6c–e, and Supplementary Fig. 5b, c and d). Taken together, the observation that knockdown of *Senp6* and *Senp7*, which belong to the same sub-class of the SENP family of isopeptidases, selectively suppresses NLRP3-dependent activation of caspase-1 strongly suggests that the NLRP3 inflammasome is regulated in a SUMO-dependent fashion.

**Discussion**
The NLRP3 inflammasome plays a key role in host defence, and is activated by numerous pathogen-associated and danger-

**Fig. 2** Signal II triggers depletion of sumoylated NLRP3. **a** Nigericin treatment results in the reduction of sumoylated NLRP3. HEK293T cells expressing FLAG-NLRP3 were stimulated with nigericin for the indicated time points. FLAG-NLRP3 immunoprecipitates were analysed for sumoylation (upper panel) and ubiquitylation (middle panel). SUMO and ubiquitin smears were quantified using Image Lab Software (Bio-Rad) and the modification normalised to total immunoprecipitated NLRP3. Graphs represent fold change compared to untreated cells. The effect of nigericin stimulation on global SUMO-2/-3 levels was determined by immunoblot analysis of total cell lysates. The data depict representative immunoblots of three independent experiments.
**b** Endogenous NLRP3 interacts with SUMO-2/-3, but the interaction is lost upon NLRP3 activation by treatment with LPS and nigericin (left panel) or ATP (middle panel). Immortalised WT or *Nlrp3*^−/− murine BMDMs were left untreated, primed with LPS for 4 h or treated with LPS (4 h) followed by 30 min with nigericin or 20 min with ATP. Cells were fixed and proximity ligation assay (PLA) was performed using α-NLRP3 and either α-SUMO-2/-3 (upper panel) or α-ASC antibodies (lower panel), as indicated. Representative confocal microscopy images are shown for each condition, which was repeated at least three times, except for ATP (*n* = 1). Scale bar, 10 μm

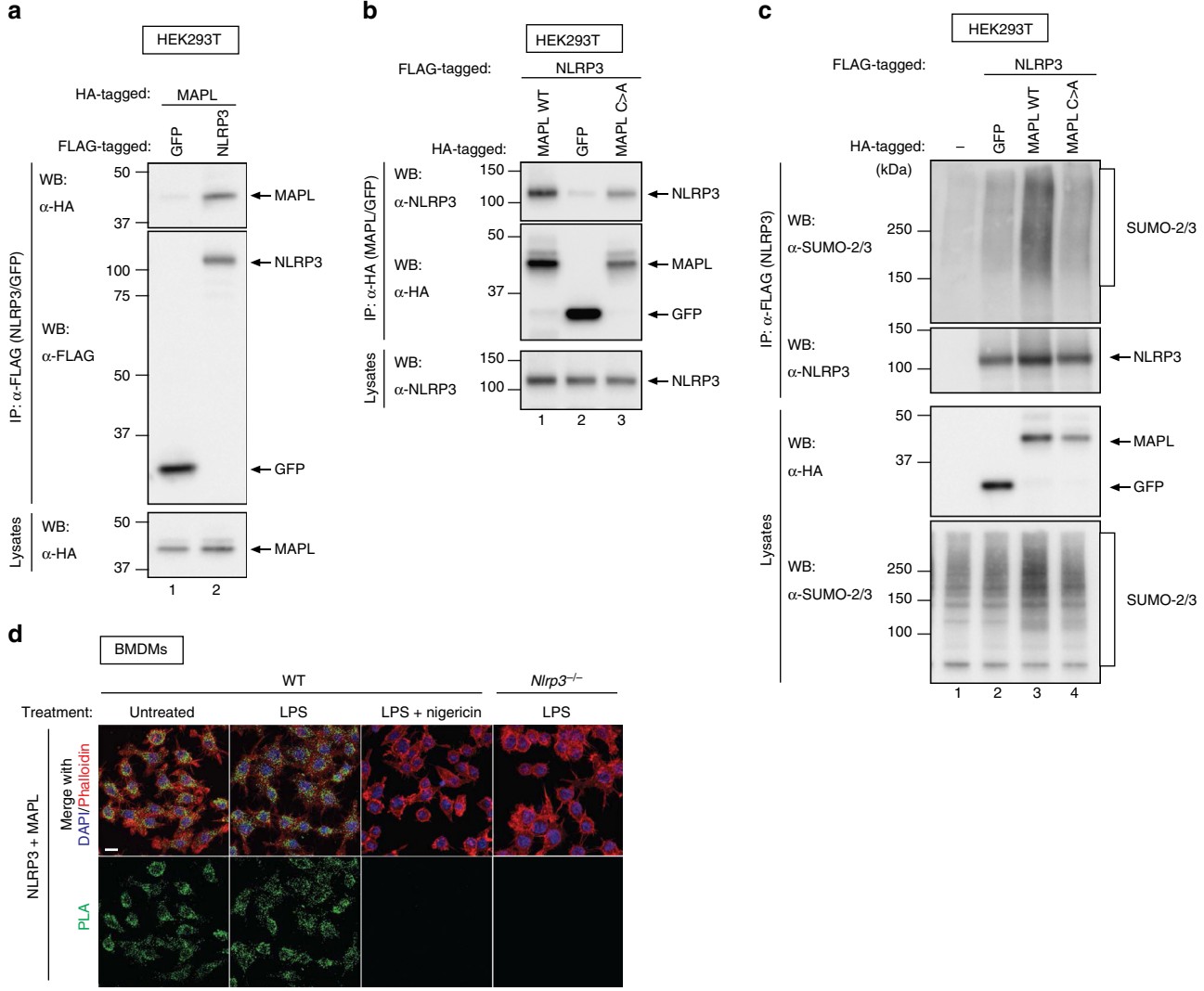

**Fig. 3** SUMO E3 ligase MAPL interacts and sumoylates NLRP3. **a**, **b** Reciprocal binding study with NLRP3 and MAPL in HEK293T. Expression and co-precipitation was determined by western blot with the indicated antibodies. **c** Sumoylation of FLAG-NLRP3 in the presence of the indicated proteins. FLAG-NLRP3 was purified under denaturing conditions and the presence of SUMO modification on NLRP3 was determined by immunoblot analysis of the eluate. Representative immunoblots of three independent experiments are shown. **d** The endogenous proximity between NLRP3 and MAPL is lost upon NLRP3 activation. Immortalized BMDMs were left untreated, primed with LPS for 4 h or treated with LPS (4 h) followed by 30 min with nigericin. PLA was performed in biological triplicate using α-NLRP3 and α-MAPL antibodies. Representative confocal microscopy images are shown for each condition. Scale bar, 10 μm

associated molecular patterns[1,44]. Several danger signals such as ATP, alum hydroxide, silica crystals, urea crystals, nigericin, and infections with bacteria, viruses, and fungi activate the NLRP3 inflammasome. How NLRP3 recognises these different ligands and whether a common signal converges downstream of PAMPs and DAMPs to activate NLRP3 has been a longstanding question in the field. Here, we show that NLRP3 is sumoylated, and that NLRP3-activating stimuli result in the removal of SUMO from NLRP3. Several lines of evidence support the notion that NLRP3 is regulated by sumoylation. First, under steady-state conditions NLRP3 associates with the SUMO E3 ligase MAPL and is sumoylated. Second, upon activation of the NLRP3 inflammasome the interaction between NLRP3 and MAPL is disrupted and NLRP3 is de-sumoylated. Third, depletion of MAPL results in enhanced inflammasome formation and activity. Fourth, an NLRP3$^{6K>R}$ mutant that is less sumoylated is significantly more potent in activating caspase-1. Fifth, NLRP3$^{K689R}$, which carries a mutation of the evolutionarily conserved SUMO acceptor K689, and cannot be sumoylated in vitro in an UBC9-

dependent manner, is hyperactive in driving caspase-1 cleavage. Finally, reduced de-sumoylation of NLRP3 through the deletion of SENPs significantly impairs its ability to drive ASC oligomerisation, caspase-1 activation and production of biological active IL-1β.

While the desumoylase enzymes SENP6 and SENP7 reverse SUMO-mediated inactivation of NLRP3, MAPL-mediated sumoylation of NLRP3 suppresses inflammasome formation. Accordingly, depletion of the SUMO E3 ligase MAPL enhances caspase-1 activation and the release of bioactive IL-1β in response to NLRP3 inflammasome triggers. Conversely, depletion of the SENP6 and SENP7 impairs inflammasome formation, caspase-1 activation and IL-1β release. MAPL-mediated regulation of IL-1β production is specific to the NLRP3 inflammasome as depletion of MAPL, SENP6 or SENP7 has no effect on the activity of the AIM2 inflammasome. Consistent with the notion that sumoylation of NLRP3 inhibits inflammasome activation, we find that treatment with nigericin and ATP results in loss of SUMO conjugation to NLRP3. Unlike other PTMs, SUMO appears to

regulate protein function through a constant cycle of SUMO-conjugation and de-conjugation. Therefore, SUMO-modification may be removed from proteins either by activating desumoylase enzymes or inhibiting SUMO-conjugation, under which case constitutively active SUMO peptidases would 'clean' such proteins of SUMO adducts. Intriguingly, *Listeria monocytogenes* and *Streptococcus pneumonia* reportedly impair host-sumoylation during infection by degrading the SUMO E2 ligase, UBC9[45]. While UBC9 degradation may be beneficial for efficient infection,

pathogen-induced depletion of UBC9 may be recognised through the concomitant accumulation of non-sumoylated NLRP3, which is known to be essential for inflammasome activation in response to both *L. monocytogenes*[11] and *S. pneumonia*[46].

In addition to sumoylation, NLRP3 is also regulated in a ubiquitin-dependent, nitric oxide-dependent and phospho-dependent manner, emphasising the importance of NLRP3 regulation[15,47–51]. In unstimulated cells, NLRP3 is inhibited by K63-linked and K48-linked poly-ubiquitylation, and de-

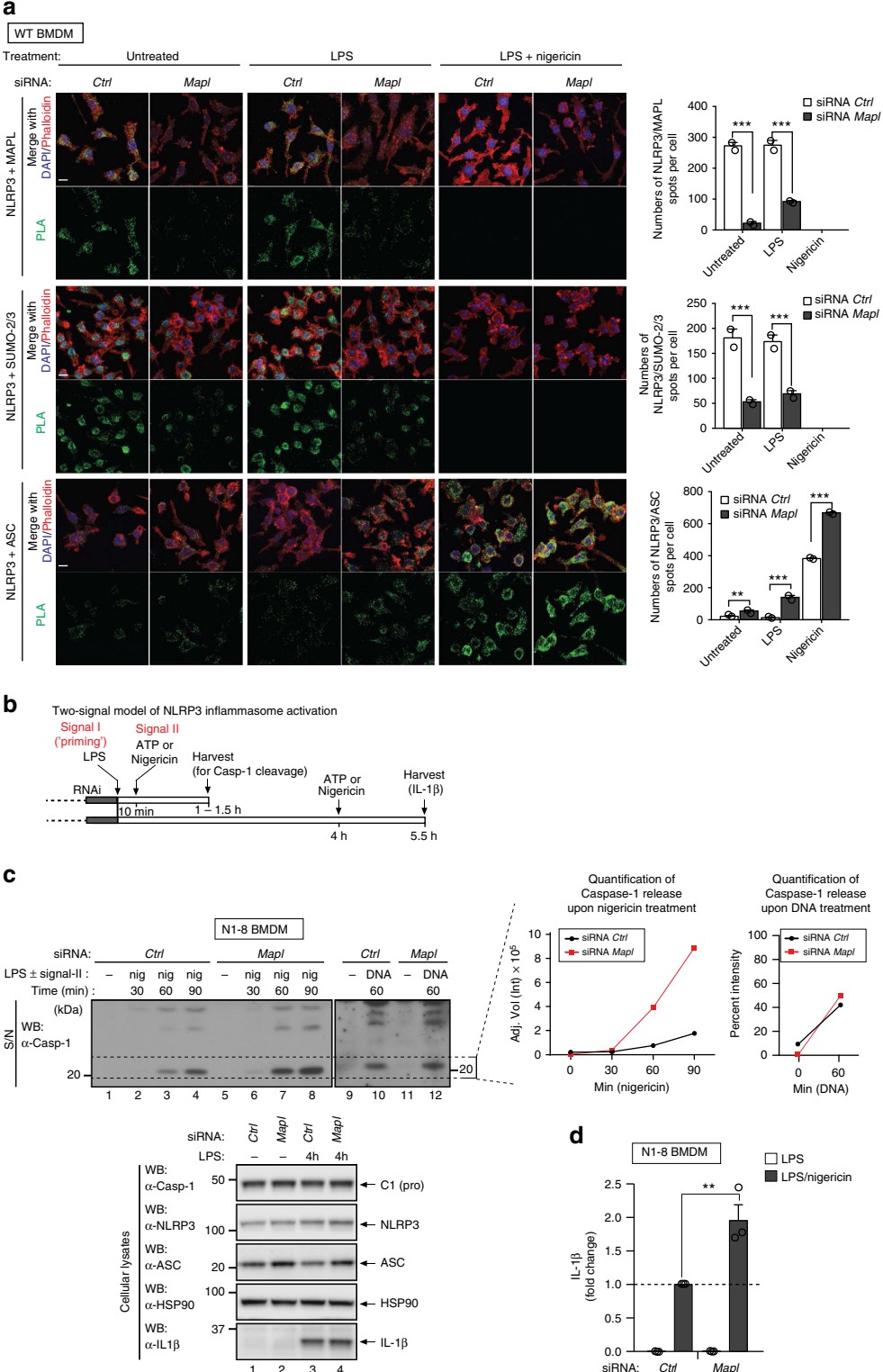

ubiquitylated by BRCC3 upon stimulation[47]. Furthermore, a number of Ub E3 ligases have been identified that promote K48-linked poly-ubiquitylation and degradation of NLRP3, such as ARIH2, FBXL2, MARCH7 and TRIM31[43,52–54]. Given that NLRP3 is inhibited by both SUMO and ubiquitin this raises the intriguing possibility of interplay between these two PTMs. One possibility is that SUMO and ubiquitin work synergistically to keep NLRP3 in an inactive configuration, or by altering the subcellular localisation of NLRP3, preventing association with the adapter ASC. According to the latter hypothesis, we noticed that PLA detects sumoylated NLRP3 in the nucleus. Thus, sumoylated NLRP3 might be shuttled into the nucleus by SUMO-binding proteins. Further, NLRP3 may be antagonistically modified with either SUMO or ubiquitin, as has previously been observed for NF-kB[55]. Recently, K689 within NLRP3 has been postulated to be a target for ubiquitylation in vitro[43]. Our study demonstrates that K689, which is located within a bioinformatically predicted SUMO consensus motif, is targeted by SUMO in an UBC9-dependent manner. This suggests that modification at this residue is crucial to prevent aberrant activation of NLRP3. Intriguingly, the defect in sumoylation of NLRP3 at K689R (NLRP3$^{K689R}$) phenocopies reported mutations identified in CAPS patients (NLRP3$^{E690K}$ and NLRP3$^{E692K}$; Infevers registry). Both these mutations cause constitutive inflammasome activation and IL-1β secretion[56–58]. As both map to the predicted negatively charged amino acid-dependent SUMO motif (NDSMs) surrounding K689, an interesting possibility is that these mutants affect proper binding to UBC9, thereby causing deregulation of NLRP3 sumoylation and activation.

Uncontrolled NLRP3 inflammasome activation underlies several human diseases, including genetically inherited auto-inflammatory conditions as well as chronic-inflammatory diseases[44,59]. Thus, activation of NLRP3 needs to be tightly controlled. It is now apparent that NLRP3 activity is regulated through the combined effects of various PTMs, including sumoylation. These modifications are likely to fine-tune NLRP3 activation to enable a rapid response to diverse danger signals. Accordingly, different NLRP3 pro-inflammatory insults may result in differential responses, depending on which PTM is alleviated. The identification of a sumoylation/de-sumoylation equilibrium that regulates NLRP3, constitutes an important advance in the field of inflammation with therapeutic potential because NLRP3 plays pivotal roles in many inflammatory disorders. As de-sumoylation is critical for full NLRP3 activation, targeting SENPs may offer new opportunities for treating inflammatory diseases that are caused by aberrant activation of the NLRP3 inflammasome, such as type-II diabetes, gout, and Alzheimer's disease. Current therapies are solely based on neutralising pro-inflammatory cytokines in the circulation[60], and since aberrant inflammasome activation also results in pyroptosis,

inhibiting the removal of SUMO from NLRP3 by targeting SENPs may provide an alternative approach for the treatment of inflammatory diseases.

## Methods

**Reagents, constructs and antibodies.** Ultra-pure EK-LPS (tlrl-peklps), ATP (tlrl-atp) and nigericin (tlrl-nig) were from Invivogen and PR-619 (SI9619) from LifeSensors. The pC1-*Hs-Nlrp3*-FLAG and *pC1-Hs-Nlrp3$^{R260W}$*-FLAG vectors were kindly provided by Veit Hornung. The four 5' K > R mutations in the 6K > R mutant of NLRP3 were introduced by replacing WT fragment between internal restriction sites, KpnI and BglII with a synthetic fragment containing the mutations (GeneArt, LifeTechnologies). The additional K652R and K689R mutants were introduced by site-directed mutagenesis. To create pC1-*Hs-Nlrp3-ΔSIM*-FLAG, WT *Nlrp3* was digested with BglII and EcoRI and the fragment replaced with a synthetically generated fragment containing the mutations (GeneArt, Life-Technologies). The pEF-*HA-SUMO3* was cloned from pcDNA3-HA-SUMO3 kindly provided by Ron Hay. The pEF-*HA-SUMO3-ΔGG* construct was generated by PCR amplification from pEF-*HA-SUMO3* using primers listed in Supplementary Table 1. *Mapl* cDNA was PCR amplified from pOTB7-*Mapl* (BioCat) and cloned into pcDNA5.5-2xHA/2xSTREP using KpnI and EcoRV sites. To generate a catalytically inactive form of MAPL, cysteine 339 was mutated to an alanine using site-directed mutagenesis (primers listed in Supplementary Table 1). All constructs used in this study were verified by DNA sequencing (Eurofins Genomics). The following antibodies were used at a dilution of 1:1000 for western blot analysis: α-caspase-1 full length and p20 (Casper-1, mouse, Adipogen), α-caspase-1 full length and p20 (Bally-1, human, Adipogen), α-NLRP3 (Cyro-2, Adipogen), α-ASC (AL117, Adipogen), α-IL-1β (AF-401-NA, R&D), α-SUMO-2/-3 (BML-PW9465, Enzo), α-pan-Ub (Z0458, Dako), α-RanGAP1 (19C7, LifeTechnologies), α-HA (3F10, Roche), α-GFP (JL-8, Clontech), α-α-tubulin (T-9026, Sigma), α-HSP90 (H-114, Santa Cruz).

**Tissue culture, RNA interference and transfections.** Immortalised WT, *Nlrp3* knockout (KO) and N1-8 BMDMs and HEK293T expressing ASC and caspase-1 (AC) cell lines were kindly provided by Emad Alnemri and were described previously[15]. HEK293T cells were obtained from ATCC. All cells were cultured in Dulbecco's modified Eagle's medium (DMEM) supplemented with 10% foetal bovine serum (Gibco) and cultured at 37 °C under 10% CO$_2$. Cell lines were tested for mycoplasma using MycoAlert$^{TM}$ Mycoplasma detection kit (Lonza). siRNAs were ordered as siGENOME SMARTpools (Dharmacon). For RNAi of mouse, Senp6 an optimised pool was used [Cat: # D-062052-01 & # D-062052-03]. All siRNA experiments were performed using 50 nM siRNA. The BMDMs were electroporated using Neon Transfection System (LifeTechnologies) according to manufacture instructions (settings: 1000 V, 40 ms, 2 pulses). For DNA transfections Genejuice Transfection Reagent (Merck Millipore) and Opti-MEM (Life-Technologies) were used according to manufacture instructions.

**Inflammasome activation.** Cells were treated 48–60 h post-siRNA in 24-well plates. For short priming, cells were treated with 500 ng/ml of ultrapure LPS for 10 min prior to the addition of the activation stimuli. For longer priming cells were treated for 4 h with LPS prior to addition of activation stimuli. To stimulate the AIM2 inflammasome 1 μg/ml of pcDNA3.1 was transfected using Lipofectamine 2000 (LifeTechnologies) for 3 h. For NLRP3 inflammasome activation cells were treated with 10 μM nigericin for 90 min or with 5 mM ATP for 60 min unless otherwise stated.

**Western blot analysis.** Following inflammasome stimulations cell supernatants were collected, centrifuged at 4 °C at 1000 rpm and lysed in 6 × SDS loading dye. The cells remaining on the plate were lysed in 1× SDS loading dye, passed through 0.8 ml columns (Pierce) to shred genomic DNA. Supernatant or cellular lysates

**Fig. 4** MAPL-mediated sumoylation of NLRP3 suppresses inflammasome activation. **a** Depletion of MAPL reduces the association between NLRP3 and SUMO-2/-3 and increases the interaction between NLRP3 and ASC. Immortalized WT BMDMs were transfected with non-targeting or *Mapl*-specific siRNAs. After 48 h, cells were left untreated, primed with LPS for 4 h or treated with LPS (4 h) followed by 30 min with nigericin. PLA was performed using α-NLRP3 and either α-MAPL (upper panel), α-SUMO-2/-3 (middle panel) or α-ASC antibodies (lower panel), as indicated. Representative confocal microscopy images are shown for each condition. Scale bar, 10 μm. The number of fluorescent spots per cell was quantified for 50 cells. The average of duplicate counts is represented on the right-hand side of the microscopy images. Circles represent individual data points and error bars represent ±SD. Data were analysed using two-way ANOVA multiple comparison test **$P < 0.01$, ***$P < 0.001$. **b** Schematic diagram of the experimental procedure performed in **c**. **c** Activation and release of caspase-1 under the indicated conditions was evaluated by western blot. N1-8 macrophages were transfected with non-targeting or *Mapl*-specific siRNAs and primed with LPS (10 min for caspase-1 or 4 h for IL-1β). Cells were subsequently stimulated with nigericin or cytoplasmic DNA. Cleaved caspase-1 (p20 band) was quantified using Image Lab Software (Bio-Rad). **d** IL-1β levels in the cell supernatant were measured by enzyme-linked immunosorbent assay (ELISA). Individual data points (circles) of three independent experiments are presented as fold change compared to Ctrl N1-8 BMDMs (non-targeting siRNA). Grey bar is the mean ± SEM. **$P < 0.01$; two-tailed Student's $t$-test on log2-transformed data comparing nigericin treated BMDMs transfected with *Mapl*-specific siRNA to Ctrl BMDMs

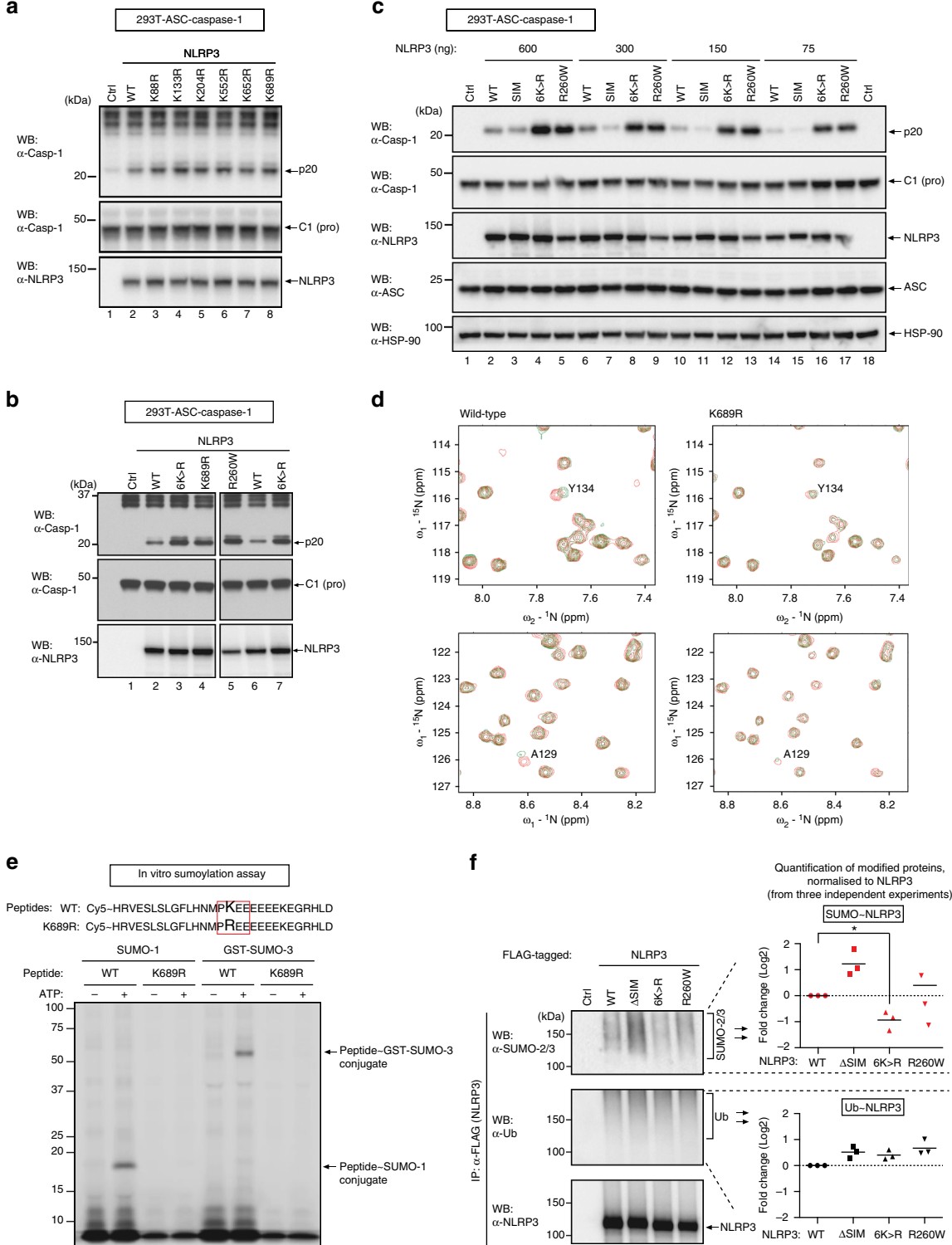

**Fig. 5** Mutation of SUMO acceptor lysines results in hyperactive NLRP3. **a–c** Immunoblot analysis of inflammasome components (caspase-1, ASC and NLRP3) in cell lysates of HEK239T cells, stably expressing ASC and caspase-1, transfected with the indicated proteins. Caspase-1 maturation (p20 band) was evaluated by western blot. **d** NMR analysis of the interaction between UBC9 and WT and K689R mutant NLRP3 peptides. Overlays of the expanded regions of $^{1}$H-$^{15}$N HSQC spectra of free UBC9 (red) and UBC9 in complex with the peptide at 8:1 molar ratio (green). Only peaks with significant chemical shift perturbation are labelled with their assignments. **e** In vitro sumoylation assays of WT and mutant peptides using SUMO-1 or GST-fusion SUMO-3. **f** HEK293T cells were transfected with plasmids expressing the indicated proteins and FLAG-NLRP3 immunoprecipitates were evaluated for sumoylation and ubiquitylation. The modifications were quantified using Image Lab Software (Bio-Rad) and normalised to total immunoprecipitated NLRP3. The data of three independent experiments are presented as log2-transformed fold changes of the mutants compared to WT NLRP3. Line represents the mean. *$P <$ 0.05; two-tailed Student's $t$-test comparing each individual mutant to WT NLRP3

samples were boiled for 5 min before separating by SDS-PAGE using NuPAGE Novex 4–12% Bis-Tris 1.0 mm 20 well pre-cast midi protein gels (Life Technologies) in MES or MOPS buffer, respectively. Uncropped western blot images are included in Supplemental Fig. 6.

**Real-time quantitative RT-PCR**. RNA was isolated using RNeasy mini kit (Qiagen) and cDNA synthesised using QuantiTech reverse transcription (Qiagen). Quantitative real-time (RT)-PCR reactions were performed using the FAM-MGB Taqman gene expression probe for SENP6 [Cat: #mm01148533_m1], SENP7 [Cat: #mm01146706_m1] and MAPL/MUL1 [Cat: #mm00503326_m1] (Applied Biosystems). Relative mRNA levels were calculated after normalisation to Actin [Cat: #mm00607939_s1] using the ΔCt method.

**Peptide synthesis**. In glass peptide synthesis vessel, 100 mg of the chlorotrityl resin (Iris Biotech GmbH) was activated in dry dichloromethane (DCM, POCH) for an hour, followed by washing and addition of 3 eq of Fmoc-Asp(tBu)-OH activated with 6eq of DIPEA (Sigma Aldrich) in dry DCM prior to loading. Reaction was carried out for 3 h (agitation) in room temperature under argon atmosphere, followed by filtration and washing with DMF. Next, resin was filtrated and washed with DMF, Fmoc-protective groups were removed with 25%PIP/DMF for 5, 5 and 25 min and the resin was washed with DMF. Next, 2 eq of Fmoc-Leu-OH was activated with 2 eq HATU (Iris Biotech GmbH) and 2 eq of collidine and coupled to the resin. Reaction was carried out for 3 h, followed by washing and Fmoc-protective group removal. Peptide chain elongation was continued in the same manner until the last amino acid. Next, last Fmoc-protective group was removed, 1 eq of Cy5-NHS (Lumiprobe) was dissolved in DMF and 5 eq of DIPEA

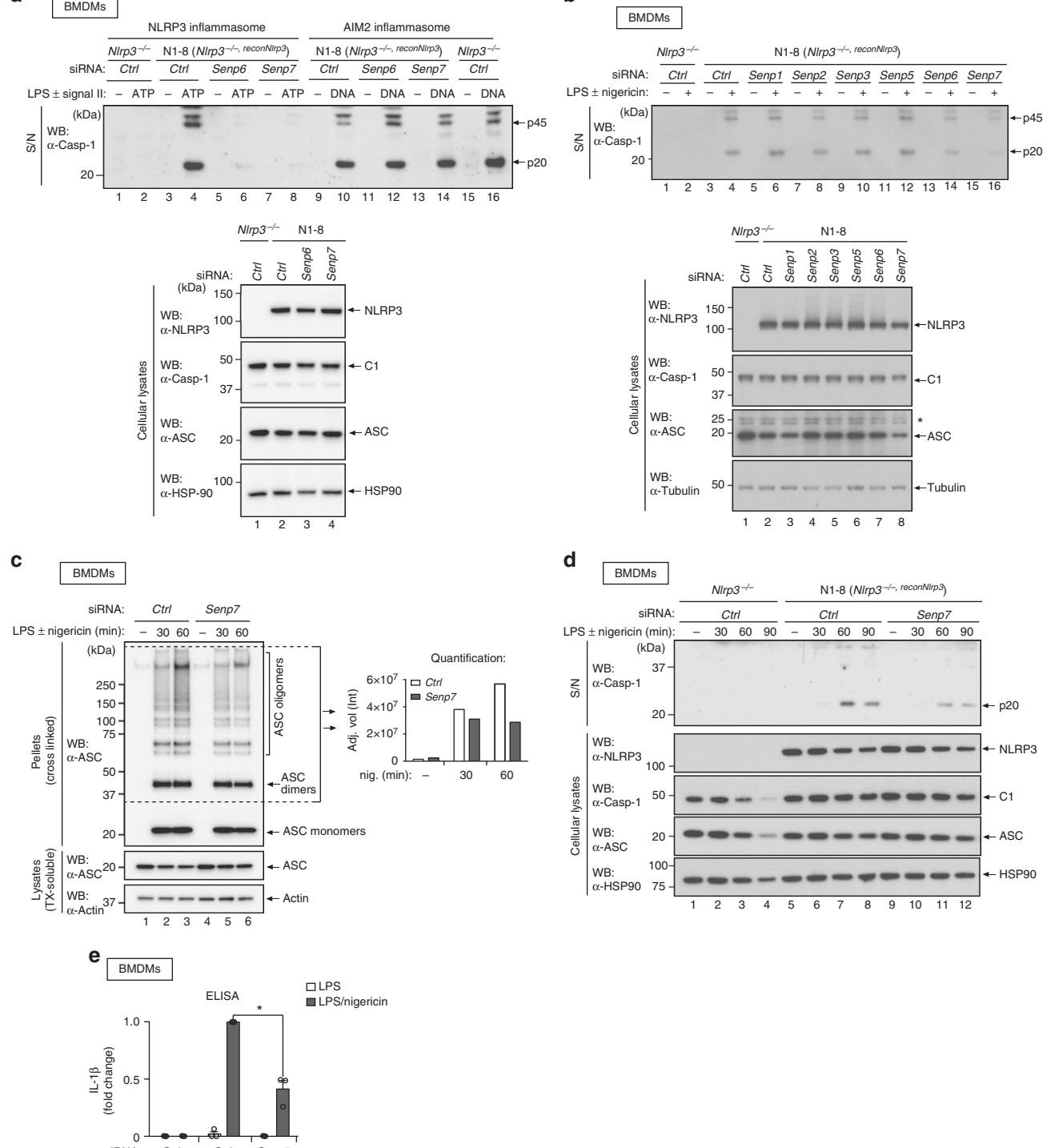

was added, followed by pouring of reaction mixture onto the resin. Reaction was carried out in room temperature for 2 h. Next, resin was washed with DMF, DCM and MeOH, dried over $P_2O_5$ for 3 h and peptides were cleaved from the resin with TFA:DCM:TIPS (44.5:44.5:1, v:v:v). Crude product was precipitated in diethyl ether, purified with HPLC (Waters, column: Speriosorb, 5 µm particle size, L x I.D > 25 cm × 4.6 mm) using a $H_2O$:acetonitrile gradient and lyophilised. Compounds were analysed with HPLC and HRMS.

**NMR studies.** [15]N-labelled UBC9 proteins were prepared at 20 µM or 50 µM in a buffer containing 100 mM sodium phosphate, pH 6.0, 5 mM dithiothreitol, 0.02% $NaN_3$, and 10% $D_2O$[42]. All NMR spectra were collected at 298 K on a Bruker Ascend 700 spectrometer. WT peptide (3.4 mM) and K689R peptide (2.4 mM) were stepwise titrated into UBC9 and chemical shift perturbation were monitored in series of the [1]H-[15]N HSQC spectra. The dissociation constant Kd was estimated using chemical shift changes as a function of the peptide:UBC9 molar ratios of the indicated residue in the proton dimension.

**In vitro sumoylation assay.** In vitro sumoylation assays were conducted by incubating the reaction mixture (30 µl) containing 0.78 µM E1, 8.4 µM peptide, 0.35 µM UBC9, 22.7 µM SUMO-1 or GST-fusion-SUMO-3, and 4 mM ATP in assay buffer (5 mM $MgCl_2$, 0.1% Tween-20, 50 mM NaCl, 20 mM HEPES pH 7.6) for 15 min at 37 °C. Two different peptides (WT, K689R mutant) were tested in the presence and absence of ATP. Reactions were quenched with 30 µl 360 mM DTT loading buffer and components were separated using SDS-PAGE. Gels were imaged using Typhoon™ 9410 Variable Mode Imager.

**Denaturing sumoylation assay HEK293T.** For sumoylation and ubiquitination assays $2 \times 10^6$ HEK293T cells were plated in 10 cm dishes and transfected with a total of 7.5 µg of DNA the following day. A modified version of "Detection of Sumoylated Proteins by Immunoprecipitation Analysis"[61] was used. Briefly, 16 h post-transfection cells were lysed in SDS lysis buffer (1% SDS, 0.15 M Tris-HCl pH 6.8 and 30% Glycerol). Lysates were boiled for 5 min and diluted 10 × in PBS-T (PBS and 1% Triton-X 100) supplemented with PR-619 (50 µM) and complete protease inhibitors. Cell lysates were sonicated for 20 s then clarified at 4 °C at 13,000 rpm for 10 min. A 20 µl of α-HA beads (SIGMA) or α-FLAG M2 beads (SIGMA) was rotated with cleared protein lysates at 4 °C for 4 h. Furthermore, 4× washes in PBS-T, supplemented with PR-619 (20 µM) and complete protease inhibitors, were performed, and samples eluted by boiling in 60 µl 1× SDS loading dye. Samples were separated by SDS-PAGE using Bolt 8% Bis-Tris plus 1.0 mm 12 well precast protein gels (Life Technologies) in MOPS buffer. Where necessary, the modification smears were quantified using Image Lab Software (Bio-Rad) and normalised to total immunoprecipitated NLRP3 in biological triplicate. A two-tailed Student's t-test was performed on log2-transformed fold change compared to WT NLRP3 using Prism 7, Graphpad.

**BMDM sumoylation assay.** NLRP3 sumoylation was assayed by immunoprecipitating NLRP3 from N1-8 BMDMs using α-FLAG M2 beads (SIGMA) as described previously[15]. Furthermore, 10 mM N-ethylmaleimide was replaced with 20 µM PR-619, and immunoprecipitates were separated by SDS-PAGE using Bolt 8% Bis-Tris plus 1.0 mm 12 well precast protein gels (Life Technologies) in MOPS buffer.

**Caspase-1 cleavage assay.** For caspase-1 cleavage assays $1 \times 10^5$ HEK293T-ASC-caspase-1 cells were plated in 24-well plates and transfected with a total of 600 ng of DNA the following day. Cells were lysed in 80 µl of DISC lysis buffer (20 mM Tris-HCL pH 7.5, 150 mM NaCl, 2 mM EDTA, 1% Triton-X 100, 10% Glycerol) supplemented with 2% SDS. To shred genomic DNA cell lysates were passed through 0.8 ml columns (Pierce). Lysates were quantified using BCA protein assay kit (Pierce) and 6× SDS loading dye added. Samples were boiled for 5 min before separating by SDS-PAGE using NuPAGE Novex 4–12% Bis-Tris 1.0 mm 20 well

precast midi protein gels (Life Technologies) in MES buffer. Caspase-1 p20 antibody (Bally-1, human, Adipogen) was used to detect total and cleaved caspase-1 in cell lysate.

**MAPL immunoprecipitation.** HEK293T cells were transfected with the indicated plasmids. Approximately 24 h after transfection, cells were lysed in 20 mM Tris pH 7.5, 150 mM NaCl, 1% Triton X-100, 10% glycerol and 2 mM EDTA containing complete protease inhibitor cocktail (Roche). Immunoprecipitations were performed using α-FLAG or α-HA agarose beads (Sigma) overnight. Samples were eluted by addition of 0.2 M glycine, separated by SDS-PAGE and examined by Western blot analysis.

**ASC oligomerization.** ASC oligomerization was performed as described in[59] with minor adaptions. BMDMs were seeded at $1 \times 10^6$ cells/ml in six-well plates 48-post siRNA treatment. Following short LPS priming and nigericin stimulation the cells were washed in ice-cold PBS and lysed in 500 µl of ice-cold buffer (20 mM HEPES-KOH, pH 7.5, 150 mM KCL, 1% NP-40, HALT protease inhibitor) by shearing times through a 21-gauge needle. Approximately, 50 µl of lysate was removed for western blot analysis. The remaining lysates was centrifuged at 6000 rpm for 10 min at 4 °C. The pellets were washed in ice-cold PBS and resuspended in 500 µl of PBS. Furthermore, 2 mM disuccinimidyl suberate (DSS) was added to the resuspended pellets, which were incubated at room temperature for 30 min with rotation. Samples were then centrifuged at 6000 rpm for 10 min at 4 °C. The supernatant was removed, and the cross-linked pellets were resuspended in 30 µl of Laemmli sample buffer. Samples were boiled for 5 min at 99 °C and analysed by western blotting.

**Confocal microscopy and PLA.** For immunofluorescence staining, $1.5 \times 10^5$ cells were plated on 13 mm glass cover slips (VWR). Cells were either left untreated, or treated with LPS (500 ng/ml) for 4 h followed or not by 10 µM nigericin for 30 min or 5 mM ATP for 20 min, before being fixed in 4% paraformaldehyde for 10 min. Following 10 min permeabilization with PBS containing 0.5% Triton X-100, cells were incubated for 1 h in 5% BSA in PBS. Respective primary antibodies were then added as follows: mouse α-NLRP3 (AG-20B-0014, AdipoGen, 1:1000), rabbit α-SUMO-2/3 (4971, Cell Signalling, 1:1000), rabbit α-ASC (AG-25B-0006, Adipo-Gen, 1:200) and rabbit α-MAPL (ab84067, Abcam, 1:1000). Alexa Fluor 633-conjugated Phalloidin (Invitrogen, 1:500) was used to stain actin filaments. Secondary fluorescent conjugated antibodies were then added as follows: 1:1000 CF488A-donkey α-mouse and 1:1000 CF488-donkey α-rabbit. Nuclei were stained with DAPI (Invitrogen). Cells were visualised by confocal microscopy (objective× 40, Zeiss LSM710). PLA was performed according to the manufacturer's protocol using the Duolink Detection Kit (Cambridge BioScience Ltd). Immunofluorescence staining of NLRP3, SUMO-2/3, ASC, and MAPL (same antibodies as above, all 1:50) for the Duolink was carried out following the above-described protocol for immunofluorescence detection up until the primary antibody incubation step. Probe incubation, ligation and amplification reaction were carried out according to the manufacturer instructions. Cy3 signal amplification was used for the assay. Cells were examined with a confocal microscope (objective× 40, Zeiss LSM 710). The number of fluorescent spots in 50 cells were quantified using ImageJ and data analysed using two-way ANOVA multiple comparison test or two-tailed Student's t-test using Prism 7, Graphpad.

**IL-1β ELISA.** The amount of IL-1β present the in the culture supernatant following inflammasome activation using long LPS priming was analysed using the Mouse IL-1β ELISA Ready-SET-Go! Kit (eBioscience) according to the manufacturer's instructions. The fold change compared to control (non-targeting siRNA) was calculated for three individual experiments and two-tailed Student's t-test was performed on the log2-transformed data using Prism 7, Graphpad.

**Fig. 6** Depletion of SENP6 and SENP7 reduces NLRP3 inflammasome activity. **a** The indicated BMDMs were transfected with non-targeting, *Senp6* or *Senp7*-specific siRNAs and primed with LPS (10 min). Cells were subsequently stimulated with ATP (1 h) or cytoplasmic DNA (3 h). Representative immunoblot analysis of cleaved caspase-1 in cell supernatant (S/N) and ASC, caspase-1 and NLRP3 in the cellular lysate. **b** The indicated BMDMs were transfected with the respective siRNAs. Cells were primed with LPS (10 min) and stimulated with nigericin (90 min). Representative immunoblot analysis of cleaved caspase-1 in cell supernatant (S/N) and ASC, caspase-1 and NLRP3 in the cellular lysate. **c** ASC oligomerisation assay. BMDMs were treated with the indicated agents. Cells were lysed and insoluble pellets cross-linked with Sulfo-DSS. Representative immunoblot analysis of ASC present in the insoluble pellet (pellets) and soluble fraction (lysates). Graph depicts the quantification of ASC oligomerisation (region indicated by dotted lines). **d** The indicated BMDMs were treated with nigericin. Immunoblot analysis of cleaved caspase-1 in the cell supernatant (S/N) and ASC, caspase-1 and NLRP3 in the cellular lysate. **e** The indicated BMDMs were treated with LPS for 4 h and stimulated with nigericin (90 min). IL-1β levels in the cell supernatant were measured by ELISA. Individual data points of three independent experiments are presented as fold change compared to Ctrl N1-8 BMDMs (non-targeting siRNA). Grey bar represents mean ± SEM. *P < 0.05; two-tailed Student's t-test on log2-transformed data comparing nigericin treated BMDMs transfected with *Senp7*-specific siRNA to Ctrl BMDMs

**Bioinformatic analysis**. Putative SUMOylation sites in NLRP3 were identified using GPS-SUMO[26], SUMOplot analysis (Abgent) software, JASSA[25] and Ron Hay's SUMO consensus motif search tool (www.lifesci.dundee.ac.uk/groups/ron_hay/pages/SumomotifQuery.html). GPS-SUMO was also used to predict SUMO-interaction motifs (SIMs) in NLRP3.

**Data availability**. The authors declare that the data supporting the findings of this study are available within the article and its supplementary information files, or are available upon reasonable requests to the authors.

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

## Acknowledgements

We would like to thank Veit Hornung and Ronald Hay for reagents and advice. We also thank members of the Meier lab for helpful discussions. Work in the Meier lab is funded by Breast Cancer Now (CTR-QR14-007), Worldwide Cancer Research (14-1328), and Komen Promise (PG12220321). T.F.-A and E.A. are funded by NIH grant number AR055398. P.K. is beneficiary of START scholarship from the Foundation for Polish Science. We acknowledge NHS funding to the NIHR Biomedical Research Centre.

## Author contributions

R.B. and P.M. conceived the study, and R.B. planned and performed experiments shown in Figs. 1a–e, 2a, 5b, c, f and 6a, b, d, e, Supplementary Figs. 4c and 5a–d. R.B. and S.W.J. planned experiments shown in Figs. 2b, 3a–d, Fig. 4a–d, Figs. 5a, 6c and Supplementary Figs. 1a–c, 2, 3a–c, 4b and S.W.J. performed the experiments. G.L. performed PLA and immunofluorescence. I.J. performed qRT-PCR for Supplementary Fig. 3a. P.K. and M.D. generated the peptides for NMR studies. C.H.C., J.C. and Y.C. performed experiments shown in Fig. 5d, e and Supplementary Fig. 4a. E.A. and T.F.A provided important reagents. T.T. helped supervise the study. R.B. and P.M. designed and supervised the study and wrote the paper.

## Additional information

**Competing interests:** The authors declare no competing interests.

