## [Peer Review File · Nature Communications]

Reviewers' comments:

Reviewer #1 (Remarks to the Author):

The manuscript by Barry et al investigates the role and the effects of sumoylation of the receptor NLRP3 and the consequences for inflammasome signalling. A combination of cell-based assays, in particular proximity-induced ligation (PLA), as well as biochemistry experiments in vitro are employed. The authors find that NLRP3 is sumoylated by the SUMO E3-ligase MAPL and that this sumoylation suppresses inflammasome activation. Towards activation, NLRP3 needs to be desumoylated by the SUMO-specific proteases SENP6 and SENP7. The authors show that reduced NLRP3 sumoylation results in enhanced inflammasome activity, as measured by caspase-1 activation and IL-1 β release. Reduction of desumoylation in turn suppresses inflammasome activity.

The inflammasome remains a topic of key biomedical significance and many regulatory aspects of its formation and its dynamics are not yet understood. The research subject is therefore timely and of high interest. The findings are interesting and could in principle be considered for publication in Nat. Comm., depending on clarification of the following issues.

Major issues:

- The evaluation of the individual contributions of the 6 possible sumoylation sites on NLRP3 to the regulation of NLRP3 activity remains incomplete. As it is reported, NLRP3 can be sumoylated at at least several of the six candidate sites. The assays of Figure 5A and 5F need therefore to be done for single site deletion mutants of each of these candidate sites, not only for K689R as it is done. The experiments should be reported and the results discussed towards the functional mechanism. The text actually seems to state such systematic experiments were done (p 10), but the data are not shown.
- The activation of the inflammasome represents a significant cellular decision that is difficult to revert and eventually leads to pyroptotic cell death. The NLRP3 (and other) inflammasomes typically manifests as a single (or sometimes 2) ASC speck(s) in the cytoplasm. The current manuscript reports such an observation of specks in Supplementary Figure S1. However, in the PLA data of ASC-NLRP3 in Figures 2 and 4, hundreds of colocalization spots are observed upon inflammasome activation. How do the authors interpret these findings and how are they in agreement with current mechanistic models of the ASC inflammasome? How do these PLA spots actually co-localize with ASC specks? This non-canonical observation requires clarification and explanation and, depending on the suggested reasons, additional experiments.
- The ASC inflammasome is generally known to serve as an amplification mechanism that by ASC polymerization amplifies the signal from single activating "seeds" towards a bulk response. Typically, the entire ASC of a cell is recruited to a single spot, once a seed has formed and forms a cross-branched, macroscopic aggregate. The authors should explain how it can be in Figure 4 that upon using mapl siRNA, some NLRP3-ASC PLA spots are formed, but these do not lead to full activation of an ASC speck.
- The authors propose that the R260W mutant escapes sumoylation control. The experiments with SENP6 SENP7 depletion should be done with this mutant to validate this.
- The structural model in Figure 5b repeats the information of Figures 1a,b, but on a structural homology model. It would be worthwhile to show this directly in Figure 1.
- The solution NMR studies and structural model shown in Figures 5c-d are non-productive towards the rest of the manuscript. Firstly, by solution NMR, the authors aim to show that a peptide with the amino acid sequence of NLRP3 motif around K689 binds to ligase UBC9 in vitro, while the control with a Lys->Arg mutation does not bind any more. On the one hand, the requirement of

the Lysine residue for sumoylation is already clear from the general knowledge of sumoylation (see introduction, multiple literature, and the prediction of sites). On the other hand, the NMR data are incorrectly analysed. When examining the chemical shifts carefully, the peptide mutated to Lysine actually does lead to chemical shift changes of residue A129, by around 0.5 line width and in the same direction as the wildtype peptide. This shift perturbation is definitively significant on modern high resolution NMR spectrometers. The correct interpretation of the data is that the mutant peptide binds albeit with a lower affinity than the wildtype. The in vitro sumoylation assay of the same peptides shown in Figure 5e is much more relevant than the NMR data, as it shows that the mutation results in inefficient sumoylation. Then, the structural model of the UBC9-peptide complex in Figure 5d is essentially a reproduction of work published previously. Its presence in the current manuscript is in so far misleading, as it seems to suggest that this structure of the complex would have been determined as part of this work. Overall, I do not really understand why the NMR data and the structural model are part of the present manuscript.

Minor issues:

- The color usage of red / grey in figure 1B is undefined. Furthermore, the text mentions that 5 of the SUMOylation site candidates are conserved. Which are these?
- Supplementary Figure S1b is not referenced nor discussed.
- The NLRP3-sumo PLA localizes to the nucleus (Figure 4). The reasons for this localization should be elucidated and discussed.

Reviewer #2 (Remarks to the Author):

Barry and colleagues study the Sumoylation of NLRP3. They find that this occurs on a number of residues, identify the E3 ligase as MAPL, and the specific SUMO peptidases for this as SENP6 and SENP7. Activation of NLRP3 requires its de-Sumoylation by these peptidases, and can be potentiated by deletion of MAPL. Overall this is a thorough and compelling series of experiments.

Major points:

- 1) Nigericin is the only NLRP3 activator tested. Just in case there is something specific to this trigger, consider some control experiments with ATP or an amyloid/crystal/particle for example.
- 2) Please define the SIM in greater detail. Location, sequence, conservation etc.
- 3) Please functionally characterize the SIM in greater detail. What effect does mutation have on inflammasome activation? Does it mediate an intramolecular SUMO interaction, or does it actually facilitate binding to SUMO on a neighboring NLRP3 subunit in the inflammasome oligomer?
- 4) Is it concerning that none of the potential SUMO sites are listed as mutants on the INFEVERS database, as being associated with CAPS? Perhaps the SIM site is? It would be nice to show that E690K or E692K have an effect on Sumoylation.
- 5) Does priming the inflammasome with LPS or GM-CSF (inducing NLRP3 expression), influence Sumoylation?

Minor points:

- 1) The difference in Figure 6A is much more significant than 6C-E, why is this?
- 2) Please present data for absolute cytokine concentrations in Fig 6E

Response to reviewer's comments on ms NCOMMS-18-01931

Please find below a point-by-point response to the reviewers' comments, with the **reviewers comments** **in blue boxes** and our response in 'plain text'.

For convenience, we have repeated the relevant new data for each reviewer:

Response to Reviewers' Comments:

Reviewer #1:

The manuscript by Barry et al investigates the role and the effects of sumoylation of the receptor NLRP3 and the consequences for inflammasome signalling. A combination of cell-based assays, in particular proximity-induced ligation (PLA), as well as biochemistry experiments in vitro are employed. The authors find that NLRP3 is sumoylated by the SUMO E3-ligase MAPL and that this sumoylation suppresses inflammasome activation. Towards activation, NLRP3 needs to be desumoylated by the SUMO-specific proteases SENP6 and SENP7. The authors show that reduced NLRP3 sumoylation results in enhanced inflammasome activity, as measured by caspase-1 activation and IL-1 β release. Reduction of desumoylation in turn suppresses inflammasome activity.

The inflammasome remains a topic of key biomedical significance and many regulatory aspects of its formation and its dynamics are not yet understood. The research subject is therefore timely and of high interest. The findings are interesting and could in principle be considered for publication in Nat. Comm., depending on clarification of the following issues.

Major issues:

The evaluation of the individual contributions of the 6 possible sumoylation sites on NLRP3 to the regulation of NLRP3 activity remains incomplete. As it is reported, NLRP3 can be sumoylated at least several of the six candidate sites. The assays of Fig. 5A and 5F need therefore to be done for single site deletion mutants of each of these candidate sites, not only for K689R as it is done. The experiments should be reported and the results discussed towards the functional mechanism. The text actually seems to state such systematic experiments were done (p 10), but the data are not shown.

We have followed the reviewers suggestion and now include additional data on the single point mutants of NLRP3, as requested.

We found that mutating individual SUMO conjugation consensus sites in isolation slightly enhanced the ability of NLRP3 to activate caspase-1, ranging from 1.3 to 1.7 fold when compared to WT NLRP3 (new Fig. 5a).

With regards to NLRP3 sumoylation, we found that mutating individual SUMO conjugation consensus sites in isolation did not significantly affect the overall smearing pattern of NLRP3 (new Suppl. Fig. 4c), which is consistent with the notion that NLRP3 is sumoylated at multiple sites. In contrast, substitution of all six K residues to R (NLRP3^{6K>R}) diminishes NLRP3 sumoylation by 50% when compared to WT NLRP3 (Fig. 5f).

We have included these data in our revised ms.

New Fig. 5a

New Suppl. Fig. 4c

The activation of the inflammasome represents a significant cellular decision that is difficult to revert and eventually leads to pyroptotic cell death. The NLRP3 (and other) inflammasomes typically manifests as a single (or sometimes 2) ASC speck(s) in the cytoplasm. The current manuscript reports such an observation of specks in Supplementary Fig. S1. However, in the PLA data of ASC-NLRP3 in Fig.s 2 and 4, hundreds of colocalization spots are observed upon inflammasome activation. How do the authors interpret these findings and how are they in agreement with current mechanistic models of the ASC inflammasome? How do these PLA spots actually co-localize with ASC specks? This non-canonical observation requires clarification and explanation and, depending on the suggested reasons, additional experiments.

PLA has previously been used to detect inflammasome formation (Li et al., 2017; Misawa et al., 2013). Moreover, a dedicated Method paper (Wu and Lai, 2016) describes the use of PLA to measure NLRP3 inflammasome formation. In these articles, the authors also observe hundreds of co-localization spots, which is consistent with our own findings. Because PLA is a PCR-based method, it is probably more sensitive than conventional antibody-based confocal microscopy. Further, it is possible that PLA detects pre-speck complexes. To avoid any confusion, we have expanded our ms to indicate this point.

For convenience, we have attached an example from the *Nat. Communications* paper:

Fig. 2b from Li et al., Nat Comms 2017

Please see:

Li, X., Thome, S., Ma, X., Amrute-Nayak, M., Finigan, A., Kitt, L., Masters, L., James, J.R., Shi, Y., Meng, G., *et al.* (2017). MARK4 regulates NLRP3 positioning and inflammasome activation through a microtubule-dependent mechanism. *Nature communications* 8, 15986.

Misawa, T., Takahama, M., Kozaki, T., Lee, H., Zou, J., Saitoh, T., and Akira, S. (2013). Microtubule-driven spatial arrangement of mitochondria promotes activation of the NLRP3 inflammasome. *Nat Immunol* 14, 454-460.

Wu, Y.H., and Lai, M.Z. (2016). Measuring NLR Oligomerization V: In Situ Proximity Ligation Assay. *Methods Mol Biol* 1417, 185-195.

We have included these citations.

The ASC inflammasome is generally known to serve as an amplification mechanism that by ASC polymerization amplifies the signal from single activating "seeds" towards a bulk response. Typically, the entire ASC of a cell is recruited to a single spot, once a seed has formed and forms a cross-branched, macroscopic aggregate. The authors should explain how it can be in Fig. 4 that upon using mapl siRNA, some NLRP3-ASC PLA spots are formed, but these do not lead to full activation of an ASC speck.

See our response above.

The PLA method seems to be more sensitive than conventional antibody-based confocal microscopy and hence also identifies ASC-NLRP3 oligomers that are not yet part of the ASC speck. It is important to note that ASC-NLRP3 oligomerisation is not synonymous with caspase-1 activation. Thus, depletion of MAPL results in a primed, pre-activation state. But for NLRP3-mediated activation of caspase-1, Nigericin or ATP is required.

Therefore, PLA can detect a primed state.

The authors propose that the R260W mutant escapes sumoylation control. The experiments with SNP6 SENP7 depletion should be done with this mutant to validate this.

Unfortunately, we were unable to generate BMDM clones expressing NLRP3^{R260W}, most likely because this gain-of-function mutant is toxic to BMDMs. Hence, we were unable to conduct the suggested experiment. Since we used this mutant only as a control, we have toned down the text regarding R260W.

The structural model in Fig. 5b repeats the information of Fig.s 1a,b, but on a structural homology model. It would be worthwhile to show this directly in Fig. 1.

We have followed the reviewer's recommendation and have moved the structural model (former Fig. 5b) to Fig. 1.

- The solution NMR studies and structural model shown in Fig.s 5c-d are non-productive towards the rest of the manuscript. Firstly, by solution NMR, the authors aim to show that a peptide with the amino acid sequence of NLRP3 motif around K689 binds to ligase UBC9 in vitro, while the control with a Lys->Arg mutation does not bind any more. On the one hand, the requirement of the Lysine residue for sumoylation is already clear from the general knowledge of sumoylation (see introduction, multiple literature, and the prediction of sites).

We are happy to move the NMR studies to the Supplementary data, should Reviewer 1 and the editor insist on this. However, we would prefer to keep this data as part of the main Figures. The peptide stretch of NLRP3 was identified based on a software prediction. Hence, we believe that it is appropriate to provide experimental evidence that this NLRP3 stretch indeed interacts with UBC9, even though this is expected. As indicated, we are happy to move this into Supplementary Figures should this be wished.

Please advise.

On the other hand, the NMR data are incorrectly analysed. When examining the chemical shifts carefully, the peptide mutated to Lysine actually does lead to chemical shift changes of residue A129, by around 0.5 line width and in the same direction as the wildtype peptide. This shift perturbation is definitively significant on modern high resolution NMR spectrometers. The correct interpretation of the data is that the mutant peptide binds albeit with a lower affinity than the wildtype. The in vitro sumoylation assay of the same peptides shown in Fig. 5e is much more relevant than the NMR data, as it shows that the mutation results in inefficient sumoylation.

We thank the reviewer to highlight this oversight. The reviewer is of course correct that the Lysine mutant still binds, albeit with much lower affinity than the wildtype. It is correct to state that this is not a binary interaction (yes or no), but a difference in affinity.

We have corrected the text accordingly.

Then, the structural model of the UBC9-peptide complex in Fig. 5d is essentially a reproduction of work published previously. Its presence in the current manuscript is in so far misleading, as it seems to suggest that this structure of the complex would have been determined as part of this work. Overall, I do not really understand why the NMR data and the structural model are part of the present manuscript.

The model was included for illustrative purpose only, and labelled as 'model'. We apologise if the inclusion of this model was perceived as mis-leading. We have moved the structural model of the UBC9-peptide complex into the Suppl. Fig. (Supplementary Fig. 4a). We are also happy to delete the structural model entirely should this be desired.

Please advise.

Minor issues:

- The color usage of red / grey in Fig. 1B is undefined. Furthermore, the text mentions that 5 of the SUMOylation site candidates are conserved. Which are these?

We have corrected these issues.

- We have defined the colour usage in the Fig. legend of Fig. 1b.
- The following consensus sites are evolutionarily conserved: K88, K133, K204, K652 and K689. We have amended the text to clearly state this.

Supplementary Fig. S1b is not referenced nor discussed.

This has been corrected.

The NLRP3-sumo PLA localizes to the nucleus (Fig. 4). The reasons for this localization should be elucidated and discussed

The reviewer is correct that a large portion of the NLRP3-SUMO PLA speckles appear to localize to the nucleus. We are currently perusing this observation in a separate study. Our working hypothesis is that sumoylated NLRP3 might be shuttled into the nucleus by SUMO-binding proteins were NLRP3-SUMO would be sequestered away from ASC. We have mentioned this possibility in the 'Discussion' of our revised ms. We feel that elucidating this shuttling/sequestration mechanism is beyond the scope of the current manuscript and is best dealt with by a separate investigation.

Reviewer #2 (Remarks to the Author):

Barry and colleagues study the Sumoylation of NLRP3. They find that this occurs on a number of residues, identify the E3 ligase as MAPL, and the specific SUMO peptidases for this as SENP6 and SENP7. Activation of NLRP3 requires its de-Sumoylation by these peptidases, and can be potentiated by deletion of MAPL. Overall this is a thorough and compelling series of experiments.

Major points:

1) Nigericin is the only NLRP3 activator tested. Just in case there is something specific to this trigger, consider some control experiments with ATP or an amyloid/crystal/particle for example.

We have followed the reviewers suggestion and have included ATP as a stimulus. As shown in **new Fig. 2b**, we find that treatment with ATP or Nigericin results in loss of sumoylation of NLRP3.

B BMDMs

New Fig. 2b

2) Please define the SIM in greater detail. Location, sequence, conservation etc

We have modified our text to provide more details on the putative SIM. We now state:

'...In addition to SUMO consensus motifs, we also identified a putative SUMO interaction motif (SIM) within the LRR of mouse (Q8R4B8, SIM: amino acids 797-800 (LVEL) and human NLRP3 (Q96P20, SIM: amino acids 800-803 (LVEL)). This site was the only putative SIM that was predicted by both Jassa and GPS algorithms. Although this putative SIM is evolutionarily conserved it has a relatively low probability score.'

3) Please functionally characterize the SIM in greater detail. What effect does mutation have on inflammasome activation? Does it mediate an intramolecular SUMO interaction, or does it actually facilitate binding to SUMO on a neighboring NLRP3 subunit in the inflammasome oligomer?

We have followed the reviewer's suggestion and provide additional data on the putative SIM. These data suggest that the SIM contributes to caspase-1 activation. Accordingly, mutating the SIM (LEVEL to AAEA) reproducibly lowered the ability of NLRP3 to activate caspase-1 (new Fig. 5c). The suggestion that the SIM might facilitate binding to SUMO on a neighbouring NLRP3 molecule in an inflammasome oligomer is an attractive model. However, mutating the putative SIM of NLRP3 (NLRP3^{SIM-mut}) causes enhanced sumoylation of NLRP3 (see Fig. 5f). The enhanced sumoylation, in turn, might explain why this mutant is less active. Clearly, future work will be required to tease apart the different possibilities.

We have expanded our ms to include these data and discuss the various possibilities.

New Fig. 5c

4) Is it concerning that none of the potential SUMO sites are listed as mutants on the INFEVERS database, as being associated with CAPS? Perhaps the SIM site is? It would be nice to show that E690K or E692K have an effect on Sumoylation.

We thank the reviewer to raise this point. While the actual sumoylated K residue is not in the INFEVERS database, there are two disease mutants (E690K and E692K) that map to the evolutionary conserved SUMO consensus motif surrounding K689. These NLRP3 missense mutations (E690K and E692K (Infevers registry)) result in constitutive inflammasome activation (Supplementary Fig. 4a) and IL-1 β secretion⁵⁷⁻⁵⁹ and hence phenocopy the mutation of the SUMO acceptor lysine K689R.

NLRP3 is sumoylated at multiple sites. Hence, like for the K689R mutant, mutating E690K or E692K in isolation did not abrogate sumoylation of NLRP3. In contrary, the E690K mutant seems to be more ubiquitylated and sumoylated, albeit it is also slightly more expressed (new Supplementary Fig. 4d). This may suggest that the E690K mutation affects the normal ubiquitylation and sumoylation pattern (K occupancy and/or chain extension).

We have expanded our ms to indicate these points.

New Supplementary Fig. 4a,d

5) Does priming the inflammasome with LPS or GM-CSF (inducing NLRP3 expression), influence Sumoylation?

We have followed the reviewer's suggestion and have evaluated whether LPS treatment influences sumoylation. We found that priming with LPS did not influence NLRP3 sumoylation. Accordingly, the appearance of PLA SUMO:NLRP3 speckles did not change following treatment with LPS. The data shows the PLA quantification from three independent experiments.

Rebuttal Fig. 1

Minor points:
 1) The difference in Fig. 6A is much more significant than 6C-E, why is this?

The difference is most likely due to variation in RNAi efficiency, which is variable in BMDMs. We have attempted to generate SENP KO BMDMs but unfortunately were unable to isolate individual clones as SENP deletion seem to be toxic to BMDMs. Hence, we were unable to generate more consistent genetic model systems (CRISPR).

2) Please present data for absolute cytokine concentrations in Fig 6E

We have followed the reviewer's suggestion and present both fold changes as well as the absolute cytokine concentrations. The absolute values are shown in the **Supplementary Fig. 5d**.

New Supplementary Fig. 5d

REVIEWERS' COMMENTS:

Reviewer #1 (Remarks to the Author):

The authors have carefully addressed all issues raised and adapted their manuscript accordingly. I recommend publication without further changes.

Reviewer #2 (Remarks to the Author):

My comments have been addressed.